# PSC: Posterior Sampling-Based Compression

**Noam Elata**                                                     *noamelata@campus.technion.ac.il*
*Department of Electrical and Computer Engineering*
*Technion - Israel Institute of Technology*

**Tomer Michaeli**                                                 *tomer.m@ee.technion.ac.il*
*Department of Electrical and Computer Engineering*
*Technion - Israel Institute of Technology*

**Michael Elad**                                                   *elad@cs.technion.ac.il*
*Department of Computer Science*
*Technion - Israel Institute of Technology*

**Reviewed on OpenReview:** *https://openreview.net/forum?id=OsqgU6Jz4t*

## Abstract

Diffusion models have transformed the landscape of image generation and now show remarkable potential for image compression. Most of the recent diffusion-based compression methods require training and are tailored for a specific bit-rate. In this work, we propose Posterior Sampling-based Compression (PSC) – a zero-shot compression method that leverages a pre-trained diffusion model as its sole neural network component, thus enabling the use of diverse, publicly available models without additional training. Our approach is inspired by transform coding methods, which encode the image in some pre-chosen transform domain. However, PSC constructs a transform that is adaptive to the image. This is done by employing a zero-shot diffusion-based posterior sampler so as to progressively construct the rows of the transform matrix. Each new chunk of rows is chosen to reduce the uncertainty about the image given the quantized measurements collected thus far. Importantly, the same adaptive scheme can be replicated at the decoder, thus avoiding the need to encode the transform itself. We demonstrate that even with basic quantization and entropy coding, PSC's performance is comparable to established training-based methods in terms of rate, distortion, and perceptual quality. This is while providing greater flexibility, allowing to choose at inference time any desired rate or distortion.[1]

## 1 Introduction

Diffusion models excel at generating high-quality images (Ho et al., 2020; Sohl-Dickstein et al., 2015; Song et al., 2020; Dhariwal & Nichol, 2021; Vahdat et al., 2021; Rombach et al., 2022). Their versatility has enabled solutions for diverse tasks, including inverse problems (Saharia et al., 2021; 2022; Chung et al., 2023; Kawar et al., 2021; 2022a; Song et al., 2023), image editing (Meng et al., 2021; Brooks et al., 2023; Kawar et al., 2023; Huberman-Spiegelglas et al., 2023), and uncertainty quantification (Belhasin et al., 2023; Horwitz & Hoshen, 2022). Conveniently, many of these applications can utilize pre-trained diffusion models without requiring task-specific training.

Image compression is fundamental for efficient storage and transmission of visual data, attracting significant research attention over decades. Effective compression schemes preserve essential image information while discarding less critical components, establishing a lossy compression paradigm that balances image quality against file size. Traditional compression methods like JPEG (Wallace, 1991) and JPEG2000 (Skodras et al., 2001) achieve this through fixed whitening transforms and coefficient quantization, allocating bits based on

---

[1]Code implementation for PSC is available at `https://github.com/noamelata/PSC`.

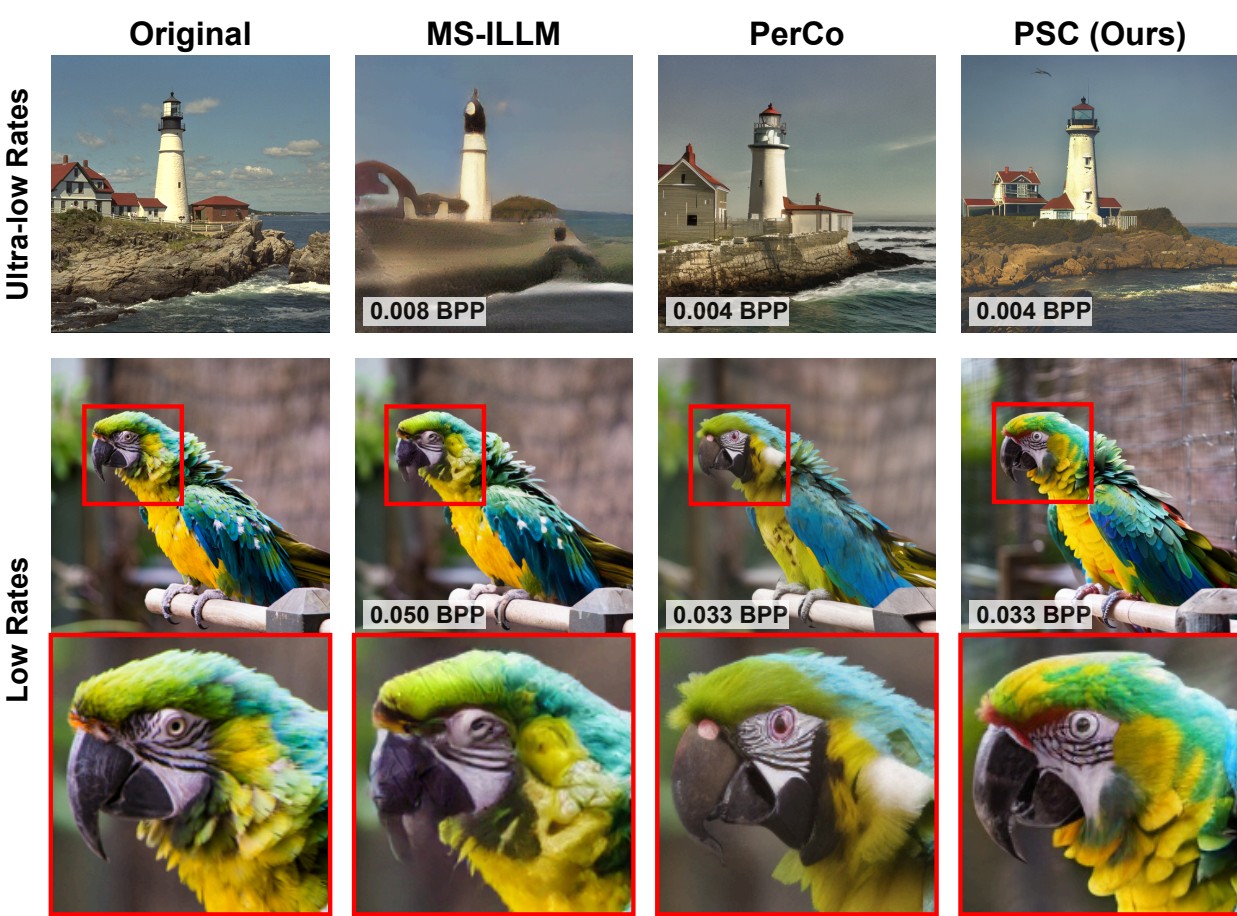

Figure 1: **Images compressed with latent-PSC at low bit-rates.** Latent-PSC leverages pre-trained latent diffusion models to deliver high perceptual quality at any compression rate, indicated by the bits-per-pixel (BPP) for each decompressed result. Top: Example at ultra-low rates demonstrates that PSC maintains high image quality while preserving the original image composition. Bottom: Example at low rates with zoomed detail highlights PSC's capacity to maintain fine details, adapting seamlessly to any compression rate without training.

the coefficients' importance and applying entropy coding for additional lossless compression. More recently, neural compression methods have demonstrated improved performance over their classical counterparts, by incorporating quantization and entropy-coding directly into their training objectives (Ballé et al., 2018; Minnen et al., 2018; Cheng et al., 2020; Ballé et al., 2016; Theis et al., 2017; Toderici et al., 2015). In this context, deep generative models, such as GANs (Mentzer et al., 2020) or diffusion models (Yang & Mandt, 2024), can improve the perceptual quality of decompressed images. While some approaches use generative modeling for post-hoc restoration of existing compression algorithms (Bahat & Michaeli, 2021; Kawar et al., 2022b; Saharia et al., 2022; Man et al., 2023; Song et al., 2023; Ghouse et al., 2023; Xu et al., 2024), their performance gains remain limited as they do not modify the encoder. Alternative methods that integrate neural compression with diffusion model-based decoding (Careil et al., 2023; Yang & Mandt, 2024; Relic et al., 2024) achieve impressive results but require rate-specific diffusion model training for a single modality, leading to reduced flexibility.

In this paper, we introduce a lossy compression method that uses a pre-trained diffusion model. This allows leveraging publicly available models without requiring additional training, and thus enables lossy compression of any data type on which a diffusion model has been trained. Our paradigm is very different

from recent neural compression literature, and can be considered an adaptive version of classical transform coding techniques. Specifically, our work draws inspiration from AdaSense (Elata et al., 2024b), which utilizes a pre-trained diffusion-model to identify an image-specific linear transform, from which the image can be reconstructed with near-minimal error. Here, we propose to use the same approach but for constructing an *image-specific compression transform*. A key challenge in doing so, is that in addition to encoding the transformed image, this approach seemingly requires encoding also the transform itself, which is completely impractical. However, our key observation is that the communication of the transform can be bypassed altogether. This allows our method to perform well at extremely low bit rates, as we illustrate in Fig. 1.

Our compression scheme, which we coin Posterior Sampling-based Compression (PSC), uses a diffusion-based posterior sampler to progressively transform and quantize the given image. In each step, the next rows of the transform matrix are chosen in a way that minimizes the uncertainty that remains about the image, given the code constructed so far. The decoder replicates these calculations (fixing the same seed) such that both the encoder and the decoder reproduce exactly the same image-adaptive transform, eliminating the need for transmitting side-information beyond the quantized coefficients of the image in the transform domain.

PSC can be applied at any rate or distortion level prescribed by the user at inference time, and enables decompression at high perceptual quality. We evaluate PSC's effectiveness against established compression methods on the ImageNet dataset (Deng et al., 2009), comparing both distortion and image quality. We also explore the compression of general images using state-of-the-art (SOTA) text-to-image latent diffusion models (Rombach et al., 2022), incorporating textual descriptions within the compressed representation. We evaluate this latent-PSC variant against high-perceptual-quality compression algorithms (Careil et al., 2023; Muckley et al., 2023) on the CLIC (Toderici et al., 2020) and DIV2K (Agustsson & Timofte, 2017) datasets. Our experiments demonstrate that PSC is competitive with established methods, despite using a pre-trained diffusion model as its sole neural network component. While our preliminary implementation employs simplified quantization, lacks tailored entropy coding and has high computational complexity, the results underscore the potential of our approach for flexible compression applications. Future advances in diffusion-based posterior samplers, combined with our training-free compression framework, have the potential to yield significant improvements in compression of images as well as other signals of interest.

To summarize, PSC offers the following advantages:

- It provides precise instance-level control over rate and distortion. This is achieved by accumulating measurements until reaching the desired rate or distortion level.

- For the same compressed code, decoding can aim for either high-quality or low-distortion. This is done by appropriately choosing the final restoration algorithm.

- As a zero-shot approach, PSC generalizes to any data type supported by diffusion models, extending beyond image-specific applications.

- Our method will inherently benefit from future advances in generative modeling and posterior sampling, without requiring modifications or re-training.

## 2 Background

### 2.1 Posterior Sampling for Linear Inverse Problems

Linear inverse problems are tasks in which a signal $\mathbf{x}$ needs to be recovered from linear measurements $\mathbf{y} = \boldsymbol{H}\mathbf{x}$, where $\boldsymbol{H} \in \mathbb{R}^{d \times D}$ is a known matrix. One approach to obtaining plausible reconstructions, is to draw samples from the posterior distribution $p(\mathbf{x}|\mathbf{y})$. Recent research demonstrates that pre-trained diffusion models, which were originally trained to sample from the prior distribution $p(\mathbf{x})$, can be adopted in a zero-shot manner to approximately sample from the posterior distribution (Kawar et al., 2022a; Chung et al., 2023; Song et al., 2023). This can be done for any vector of measurements $\mathbf{y}$ and any measurement matrix $\boldsymbol{H}$, which are provided to the sampler at inference time.

## 2.2 Adaptive Compressed Sensing

In several important problems, such as single pixel cameras and MRI, there is flexibility in designing the operator $\boldsymbol{H}$. This gives rise to a compressed sensing problem (Donoho, 2006), in which the question is which matrix $\boldsymbol{H}$ (of predefined dimensions) allows reconstructing the signals of interest with minimal error. Recently, Elata et al. (2024b) introduced AdaSense, a method that utilizes posterior samplers to progressively construct the sensing matrix $\boldsymbol{H}$ so as to optimally represent the input image.

The algorithm works as follows. At stage $k$, we have the currently held[2] matrix $\boldsymbol{H}_{0:k}$ and measurements $\mathbf{y}_{0:k} = \boldsymbol{H}_{0:k}\mathbf{x}$. The selection of the next row is done by generating samples from the posterior $p(\mathbf{x}|\mathbf{y}_{0:k}, \boldsymbol{H}_{0:k})$ using any posterior sampler and applying principal component analysis (PCA) to identify the principal direction of the uncertainty that remains in $\mathbf{x}$ given the current $\mathbf{y}_{0:k}$. This direction is chosen as the next row in $\boldsymbol{H}$, which is used to acquire a new measurement of $\mathbf{x}$. More generally, instead of selecting one new measurement at a time, it is possible to add $r$ new measurements in each iteration[3]. This selection of the new rows of $\boldsymbol{H}$ is optimal in the sense that it allows achieving the minimal possible mean squared error (MSE) that can be obtained with a linear decoder (even though a more sophisticated reconstruction method is eventually used).

A single iteration of the method is detailed in Alg. 1. The process repeats until $\boldsymbol{H}$ reaches the desired dimensions. It is important to note that AdaSense produces an *image-specific sensing matrix* $\boldsymbol{H}$ and corresponding measurements $\mathbf{y}$. These can be used for obtaining a candidate reconstruction $\hat{\mathbf{x}}$ by leveraging the final posterior, $p(\mathbf{x}|\mathbf{y}, \boldsymbol{H})$, where $\boldsymbol{H}$ is the final matrix (obtained at the last step). This final reconstruction may employ a different, more

---

**Algorithm 1** A single iterative step of row selection Denoted as SelectNewRows $(\boldsymbol{H}_{0:k}, \mathbf{y}_{0:k}, r)$

**Require:** Previous sensing rows $\boldsymbol{H}_{0:k}$, corresponding measurements $\mathbf{y}_{0:k}$, number of new measurements $r$
1: $\{\mathbf{x}_i\}_{i=1}^s \sim p(\mathbf{x}|\mathbf{y}_{0:k}, \boldsymbol{H}_{0:k})$
2: $\{\mathbf{x}_i\}_{i=1}^s \leftarrow \{\mathbf{x}_i - \frac{1}{s}\sum_{j=1}^s \mathbf{x}_j\}_{i=1}^s$
3: $\tilde{\boldsymbol{H}} \leftarrow$ Append top $r$ singular vectors of $(\mathbf{x}_1, \ldots, \mathbf{x}_s)^\top$
4: **return** $\tilde{\boldsymbol{H}}$

---

accurate (and possibly more computational demanding) posterior sampler, as it executes only once. Please see (Elata et al., 2024b) for further details.

## 2.3 Transform Coding

Image compression algorithms based on the transform coding paradigm, like the widely used JPEG (Wallace, 1991), apply a pre-chosen, fixed and orthonormal[4] transform on the input image, $\mathbf{x} \in \mathbb{R}^D$, obtaining its representation coefficients. These coefficients go through a quantization stage, in which portions of the transform coefficients are discarded entirely, and other portions are replaced by finite precision versions, with a bit-allocation that depends on their importance for the image being compressed. As some of the transform coefficients are discarded, this scheme can be effectively described as using a partial transform matrix $\boldsymbol{H} \in \mathbb{R}^{d \times D}$ with orthogonal rows, and applying the quantization function $Q(\cdot)$ to each of elements of the remaining measurements $\mathbf{y} = \boldsymbol{H}\mathbf{x}$. Image compression algorithms include an entropy coding stage that takes the created bit-stream and passes it through a lossless coding block (*e.g.* Huffman coding, arithmetic coding, etc.) for a further gain in the resulting file-size.

The decoder has knowledge of the transform used, $\boldsymbol{H}$. Therefore, given the encoded signal, $Q(\mathbf{y})$, the decoder can generate a reconstructed image *e.g.* by using $\boldsymbol{H}^\dagger Q(\mathbf{y})$, where $\boldsymbol{H}^\dagger$ is the Moore–Penrose pseudo-inverse of $\boldsymbol{H}$. In practice, more sophisticated restoration methods are often used, some of which make use of neural networks.

When a compression algorithm is said to be progressive, this means that the elements of $\mathbf{y}$ are sorted based on their importance, and transmitted in their quantized form sequentially, enabling a decompression of the image at any stage based on the received coefficients so far. Progressive compression algorithms are highly

---

[2]In our notations, the subscript $\{0 : k\}$ implies that $k$ elements are available, from index 0 to index $k - 1$.

[3]This algorithm presents a strategy of choosing the $r$ leading eigenvectors of the PCA at every stage instead of a single one, getting a substantial speedup in the measurements' collection process at a minimal cost to adaptability.

[4]Having orthogonal rows has two desirable effects – easy-inversion and a whitening effect. Using a biorthogonal system as in JPEG2000 (Skodras et al., 2001) has similar benefits.

---

**Algorithm 2** PSC Encoder

**Require:** Image $\mathbf{x}$, number of steps $N$, number of measurements per step $r$.
1: **initialize** $\mathbf{y}_{0:0}, \boldsymbol{H}_{0:0}$ as an empty vector/matrix
2: **for** $n \in \{0 : N - 1\}$ **do**
3:     $\boldsymbol{H}_{nr:nr+r} \leftarrow \text{SelectNewRows} (\boldsymbol{H}_{0:nr}, \mathbf{y}_{0:nr}, r)$
4:     $\mathbf{y}_{0:nr+r} \leftarrow \text{Append}\,[\mathbf{y}_{0:nr}, Q(\boldsymbol{H}_{nr:nr+r}\mathbf{x})]$
5:     $\boldsymbol{H}_{0:nr+r} \leftarrow \text{Append}\,[\boldsymbol{H}_{0:nr}, \boldsymbol{H}_{nr:nr+r}]$
6: **end for**
7: **return** Entropy Encode($\mathbf{y}_{0:Nr}$)

---

**Algorithm 3** PSC Decoder

**Require:** compressed representation $\mathbf{y}$, number of steps $N$, number of measurements per step $r$.
1: **initialize** $\mathbf{y}_{0:Nr} \leftarrow \text{Entropy Decode}(\mathbf{y})$
2: **initialize** $\boldsymbol{H}_{0:0}$ as an empty matrix
3: **for** $n \in \{0 : N - 1\}$ **do**
4:     $\boldsymbol{H}_{nr:nr+r} \leftarrow \text{SelectNewRows} (\boldsymbol{H}_{0:nr}, \mathbf{y}_{0:nr}, r)$
5:     $\boldsymbol{H}_{0:nr+r} \leftarrow \text{Append}\,[\boldsymbol{H}_{0:nr}, \boldsymbol{H}_{nr:nr+r}]$
6: **end for**
7: **return** $\hat{\mathbf{x}} = f(\mathbf{y}_{0:Nr}, \boldsymbol{H}_{0:Nr})$

---

desirable, since they induce a low latency in decompressing the image. Note that the progressive strategy effectively implies that the rows of $\boldsymbol{H}$ have been sorted as well based on their importance, as each row gives birth to the corresponding element in $\mathbf{y}$. Adopting this view, at step $k$ we consider the sorted portions of $\boldsymbol{H}$ and $\mathbf{y}$, denoted by $\boldsymbol{H}_{0:k} \in \mathbb{R}^{k \times D}$ and $\mathbf{y}_{0:k} = \boldsymbol{H}_{0:k}\mathbf{x} \in \mathbb{R}^k$. As the decoder gets $Q(\mathbf{y}_{0:k})$, it may produce $\boldsymbol{H}_{0:k}^{\dagger} Q(\mathbf{y}_{0:k})$ as a temporary output image. It is important to note that in this paradigm $\boldsymbol{H}$ is fixed, and therefore the sorting of its rows is determined only once. This chosen order is used for compressing every image $\mathbf{x}$.

## 3 Method

In this section, we present how our proposed compression scheme builds upon the framework of transform coding by leveraging the learned prior of diffusion models to determine an *image-specific transform*. By adapting the compressed sensing algorithm detailed in Sec. 2.1 to compression, PSC implements a progressive approach to select the most informative partial measurements, effectively reducing reconstruction error by progressively increasing the bit-stream length. We solve the conundrum created by using an image-specific transform, by demonstrating that both the encoder and the decoder can synchronize the transform without any use of side-information.

During each step of the compression phase, our system dynamically estimates the posterior probability distribution $p(\mathbf{x}|\mathbf{y}_{0:k}, \boldsymbol{H}_{0:k})$, which is conditioned on the previously extracted $\boldsymbol{H}_{0:k}$ and the corresponding quantized partial measurements $\mathbf{y}_{0:k}$. This estimation utilizes samples generated by a posterior sampler. Importantly, the zero-shot methods listed in Sec. 2.1 can solve any inverse problem of the form $\mathbf{y} = Q(\boldsymbol{H}\mathbf{x})$, enabling the utilization of pre-trained diffusion models without training. In practice, we use posterior samplers designed for linear inverse problems (without quantization). This allows using efficient samplers and leads to sufficiently accurate results.

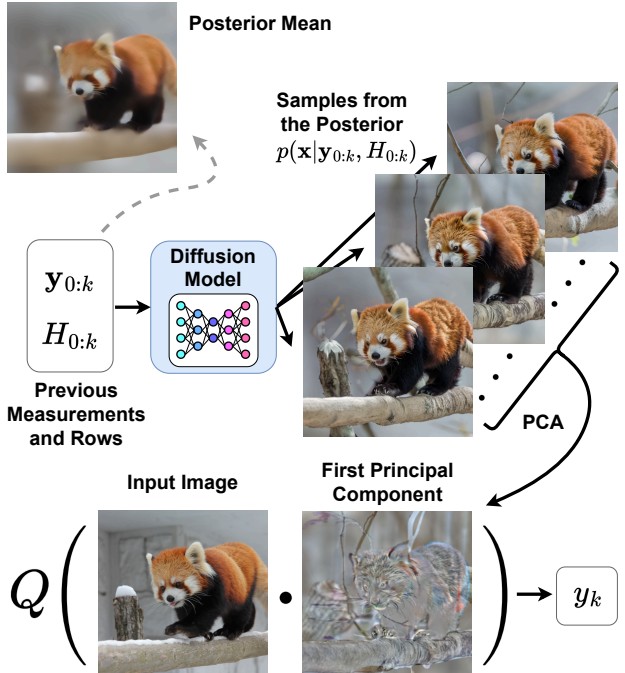

Figure 2: **A diagram of how PSC zeros in on the input image by progressively tightening the posterior.** The diagram shows a single step, where the new rows and matching measurements are computed based on the direction of largest uncertainty in the posterior distribution. The posterior mean is shown to visualize the information captured by previous iterations.

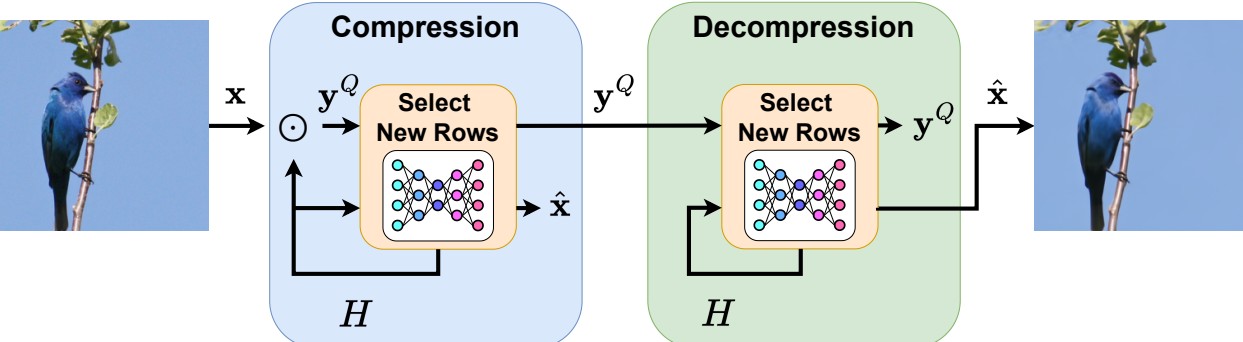

Figure 3: **PSC diagram:** Both encoder and decoder construct an image-specific transform $\boldsymbol{H}$ through an adaptive compressed sensing algorithm, progressively adding rows based on posterior sample covariance. The transmission of quantized measurements $\mathbf{y}$ ensures identical inputs at each progressive step, while a shared random seed guarantees deterministic outputs on both sides. Together, these factors enable the construction of identical transforms on both sides – eliminating the need to transmit the transform as side information.

The selection of the next row of $\boldsymbol{H}$, denoted as $\mathbf{h}_k \in \mathbb{R}^{1 \times D}$, is determined by identifying the eigenvector corresponding to the largest eigenvalue of the posterior covariance. This method ensures the projection of $\mathbf{x}$ occurs along the most informative direction, maximizing the value of incremental information gathered. The resulting measurement, $y_k = Q(\mathbf{h}_k \mathbf{x})$, is then appended to the previous compressed representation $\mathbf{y}_{0:k}$ to form $\mathbf{y}_{0:k+1}$. This process effectively reduces the uncertainty of candidate images within the posterior distribution $p(\mathbf{x}|\mathbf{y}_{0:k}, \boldsymbol{H}_{0:k})$ in a nearly optimal manner. Interestingly, as a by-product of the algorithm, the obtained sensing matrix $\boldsymbol{H}$ has orthogonal rows, disentangling the measurements, as expected from a compression algorithm. A single step of this process is described in Fig. 2.

This approach might appear counterintuitive for a transform coding, as it raises questions about the decoder's ability to determine the appropriate transform for image recovery. The naive approach of directly communicating the transform would be impractical, as it would require more bits than a lossless transmission of the image itself (the transform can be any matrix in $\mathbb{R}^{d \times D}$). However, we present an elegant solution that circumvents this challenge through our progressive compression structure, which eliminates the need to communicate the transform entirely.

More specifically, PSC maintains a synchronized and identical state between encoder and decoder throughout its operation. The system relies on an agreed-upon seed, ensuring all random sampling operations produce deterministic and reproducible outputs. PSC initiates both encoder and decoder algorithms from the same empty matrix $\boldsymbol{H}_{0:0}$ and empty vector of previous quantized projections $\mathbf{y}_{0:0}$. Assuming that the previous steps were completed successfully – i.e. the accumulated matrix $\boldsymbol{H}_{0:k}$ and quantized measurements $\mathbf{y}_{0:k}$ are identical in both the encoder and decoder, we proceed to construct the next row of $\boldsymbol{H}_{0:k}$. During this computation, all posterior samples $\{\mathbf{x}_i\}_{i=1}^{s} \sim p(\mathbf{x}|\mathbf{y}_{0:k}, \boldsymbol{H}_{0:k})$ are identical in both encoder and decoder, ensuring the synchronization of the newly computed row $\mathbf{h}_k$. The encoder then evaluates and quantizes the new measurements $y_k = Q(\mathbf{h}_k \mathbf{x})$, incorporating them into the compressed representation transmitted to the decoder. Using the compressed representation, the decoder can utilize all measurements $\mathbf{y}_{0:k+1}$ directly in subsequent steps without requiring access to the input image. This ensures that the inputs to the next iteration, $\boldsymbol{H}_{0:k+1}$ and $\mathbf{y}_{0:k+1}$, remain synchronized. This novel approach to synchronized transform reconstruction is illustrated in Fig. 3. The complete procedures for compression and decompression with PSC, including the optimization of selecting $r$ rows from matrix $\boldsymbol{H}$ for improved efficiency, are detailed in Alg. 2 and Alg. 3 respectively.

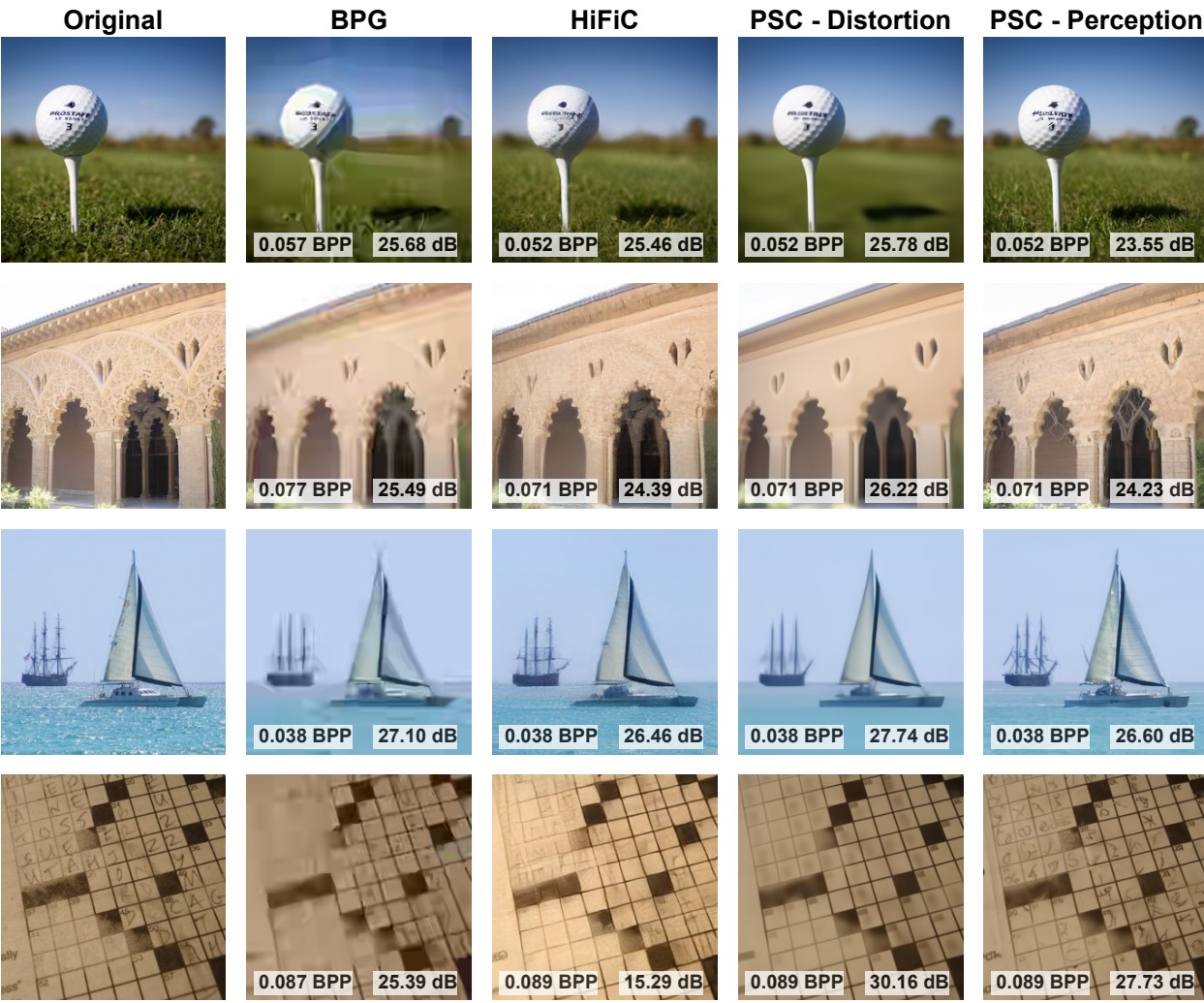

Figure 4: **Qualitative examples for compression with PSC, compared to other compression algorithms.** BPP and PSNR are reported per example. Our method can be used for both low-distortion or high perceptual quality using the same compressed representation.

## 3.1 Implementation Choices

We use DDRM (Kawar et al., 2022a) as a zero-shot posterior sampler for the selection of $\boldsymbol{H}$, due to it's relative low computational complexity. Due to the repeated sampling from different posteriors, PSC retains a high computational complexity, requiring approximately 10,000 NFEs for both compression and decompression (see runtime details in App. A.2). Nevertheless, we expect advances in diffusion models and posterior sampling to significantly expedite future versions. While the matrix $\boldsymbol{H}_{0:1}$ used in the first step can be optimized globally using images from the dataset, we keep the first step identical to the following steps as described above for simplicity. In our implementation we focus on an unsophisticated quantization approach, reducing the precision of $\mathbf{y}$ from float32 to float8 (Micikevicius et al., 2022). We employ Range Encoding implemented using (Bamler, 2022) as an entropy coding on the quantized measurements. The quantization, the posterior sampler and the entropy coding could all be improved, posing promising directions for future work. Finally, after reproducing $\boldsymbol{H}$ on the decoder side, PSC may use a different final posterior sampler during decompression, in an attempt to further boost perceptual quality for the very same measurements $\mathbf{y}$.

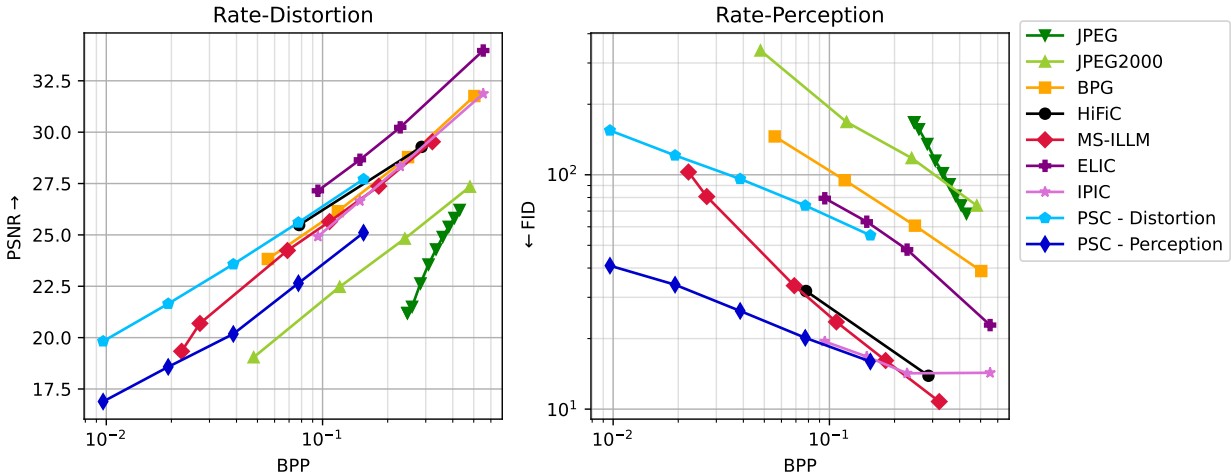

Figure 5: **Rate-Distortion (left) and Rate-Perception (right) curves for ImageNet256 compression.** Distortion is measured as average PSNR of images for the same desired rate or specified compression quality, while Perception (photorealism) is measured by FID.

## 4 Experiments

We begin with an evaluation of PSC on $256 \times 256$ color images from the ImageNet (Deng et al., 2009) dataset, using the unconditional diffusion models from (Dhariwal & Nichol, 2021) as an image prior. We compare distortion (PSNR) and bits-per-pixel (BPP) averaged on a subset of validation images, using one image from each of the 1000 classes, following (Pan et al., 2021). We apply PSC to progressively decode at higher rates, as detailed in App. A.

A key advantage of PSC is its ability to prioritize perceptual quality during decompression by changing the final reconstruction algorithm. However, this flexibility comes with a caveat: using a high-quality reconstruction algorithm will inevitably lead to higher distortion (Blau & Michaeli, 2019). Despite this, using PSC, the same compressed representation can be decoded using either a low-distortion or high perceptual quality approach with minimal additional computational cost. Specifically, we find that ΠGDM (Song et al., 2023) produces images with highest photorealism, while DDRM (Kawar et al., 2022a) leads to the lowest distortion. We present both restoration solutions as **PSC - Perception** and **PSC - Distortion** accordingly.

Figure 5 presents the rate-distortion and rate-perception curves of PSC compared to several established methods: classic compression techniques like JPEG (Wallace, 1991), JPEG2000 (Skodras et al., 2001), and BPG (Bellard, 2018). We also compare to neural compression methods, such as ELIC (He et al., 2022) and its diffusion-based derivative IPIC (Xu et al., 2024), as well as HiFiC (Mentzer et al., 2020), a prominent GAN-based neural compression method. Distortion is measured by averaging the PSNR across different algorithms for a given compression rate. Image quality is quantified using FID (Heusel et al., 2017), estimated on 50 random 128×128 crops from each image (inspired by the evaluation in (Mentzer et al., 2020)), and compared to the same set of baselines. The graphs demonstrate that PSC achieves comparable performance, particularly at low BPP regimes, when considering both distortion and image quality. Figure 4 showcases qualitative image samples compressed using different algorithms at the same rate, further supporting our findings. Notably, PSC achieves exceptional image quality despite the fact that it does not require any compression-specific training.

### 4.1 Latent PSC

Latent Text-to-Image diffusion models have gained popularity due to their ease-of-use and low computational requirements. These models employ a VAE (Kingma & Welling, 2013) to conduct the diffusion process in

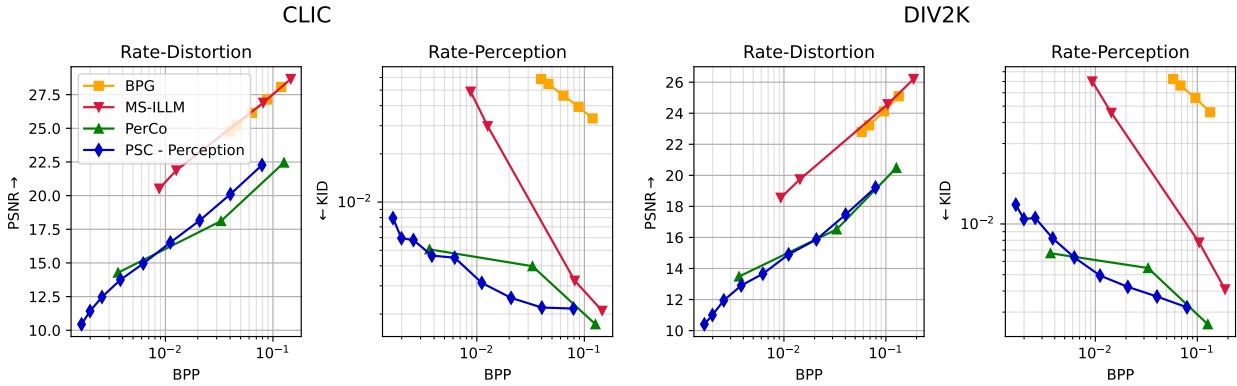

Figure 6: **Rate-Distortion and Rate-Perception curves demonstrating PSC with latent-diffusion compared to similar methods.** Left: CLIC dataset, Right: DIV2K dataset. Distortion is measured as average PSNR, while Perception (photorealism) is measured by KID.

a lower-dimensional latent space (Vahdat et al., 2021; Rombach et al., 2022). In this work we also explore the integration of PSC with Stable Diffusion (Rombach et al., 2022), a publicly available latent Text-to-Image diffusion model. This variant, named latent-PSC, operates in the latent space of the diffusion model. Both compression and decompression occur within this latent space, leveraging the model's VAE decoder to reconstruct the image from the decompressed latent representation. Additionally, we condition all posterior sampling steps on a textual description, which must be given along with the original image or inferred using an image captioning module (Vinyals et al., 2016; Li et al., 2022; 2023). The text prompt must be added to the compressed representation to avoid side-information. A detailed diagram of latent-PSC is presented in App. A.

Due to the use of the VAE decoder, we expect a significant drop in PSNR. Thus, we develop latent-PSC as an extension of **PSC - Perception**, maintaining high perceptual quality at low bit-rates. We find that the posterior sampler outlined in Nested Diffusion (Elata et al., 2024a) works best in this setting. We use images from the CLIC (Toderici et al., 2020) and DIV2K (Agustsson & Timofte, 2017) to compare Latent-PSC to PerCo (Careil et al., 2023), a recent work which also utilizes latent diffusion for low-rate image compression. Using the same base diffusion model, image captioning model, and text compression as PerCo, we demonstrate in Fig. 1 and Fig. 6 that we are comparable in terms of both distortion (PSNR) and photorealism (KID (Bińkowski et al., 2018)) on all bit-rates despite not adding any training. We also compare to MS-ILLM (Muckley et al., 2023), another compression method that focuses on high perceptual quality, as well as BPG, as a classical baseline. While the results of MS-ILLM do not suffer from the VAE-induced drop in PSNR, they do not reach the image quality of PerCo or Latent-PSC, especially at very low rates.

## 5 Related Work

Diffusion models have enhanced classical compression algorithms by providing data-driven decompression for high-perceptual quality reconstruction (Ghouse et al., 2023; Saharia et al., 2022). Several approaches implement zero-shot diffusion-based reconstruction (Kawar et al., 2022b; Song et al., 2023), offering training-free decompression, but remain constrained to specific compression algorithms, which may be found lacking. Notably, recent work by Xu et al. (2024) attempts to utilize general diffusion-based posterior samplers ro increase the image quality using a leading neural compression method. While this work employs pre-trained diffusion-based posterior samplers similar to our method, they focus on traversing the RDP trade-off (Blau & Michaeli, 2019) of existing neural compression schemes rather than developing a comprehensive compression solution.

Recent developments integrate neural compression with diffusion models for decompression. Some approaches employ separate (Hoogeboom et al., 2023) or joint (Yang & Mandt, 2024) neural compression and diffusion training to create compact representations with conditional diffusion models for high-quality decompression. Specifically, Careil et al. (2023); Relic et al. (2024) utilize latent diffusion (Rombach et al., 2022) and text-conditioned models for efficient training. Despite their promise, these methods require complex rate-specific training, limiting flexibility. While work by Gao et al. (2022) addresses this through training-free post-hoc rate reconfiguration, they incur high computational costs and performance penalties.

The concept of leveraging pre-trained diffusion models for compression originated in the DDPM publication (Ho et al., 2020), though it focused on theoretical compression limits rather than practical implementation. Similarly, Theis et al. (2022) analyzes theoretical limits using reverse channel coding techniques (Li & El Gamal, 2018), but their implementation's high computational complexity and lack of public code prevents direct comparison with our approach.

## 6 Limitation, Discussion and Conclusion

While PSC introduces a novel approach to generative model-based compression, several limitations warrant discussion. The method's primary constraint is its substantial computational overhead due to iterative diffusion model sampling. This computational burden is intrinsically linked to the zero-shot posterior sampler's quality, though ongoing advances in diffusion models and posterior sampling techniques may significantly reduce this limitation. The current implementation's simplified quantization strategy for measurements presents another constraint. Implementation of more sophisticated quantization methods could yield significant improvements in compression rates. Furthermore, PSC's restriction to linear measurements, imposed by both existing posterior sampler capabilities and the complexity of non-linear measurement optimization, indicates potential for enhancement through the exploration of non-linear measurements and corresponding inverse problem solvers. Notwithstanding these limitations, PSC represents a significant advancement in zero-shot diffusion-based image compression through several distinct advantages. The method progressively acquires informative measurements to create compressed representations, with decompression faithfully reconstructing the original image by replicating the compression algorithm's steps. Its implementation simplicity, independence from training data, and cross-domain flexibility underscore its potential impact. Future developments promise to further advance this approach to both image compression and compression in general.

## Acknowledgments

This research was partially supported by the Israel Science Foundation (ISF) under Grants 951/24, 409/24 and 2318/22, and by the Council for Higher Education – Planning and Budgeting Committee. This research was also partially supported by by a gift from Elbit Systems, and by the Ollendorff Minerva Center, ECE faculty, Technion. We would like to thank Hila Manor for her help in running evaluations, and Matan Kleiner and Sean Man for their helpful discussions and ideas.

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
