

Figure 7: **Latent-PSC diagram:** Latent Text-to-Image diffusion models such as Stable Diffusion can be used for effective image compression with PSC. The latent representation is compressed using linear measurements. The textual prompt is used for conditioning the diffusion model in both the compression and decompression, and thus this text is also transmitted.

## A    Implementation Details

For the ImageNet experiment we used the unconditional diffusion model from Dhariwal & Nichol (2021) to apply PSC. We used 25 DDRM Kawar et al. (2022a) steps to generate 16 samples, using $\eta = 1.0$ and $\eta_b = 0.0$. We added 12 rows to $\boldsymbol{H}$ in every iteration, and used matched the number of iterations to the desired rate. We restore the images using the same model with either $\Pi$GDM Song et al. (2023) with 100 denoising steps and default hyperparameters for high perceptual quality restoration, or an average of 64 DDRM Kawar et al. (2022a) samples which where produced as detailed above for low-distortion restoration. We use the Range Encoder from constriction Bamler (2022) as an entropy encoder[5] in all our experiments.

In the latent diffusion experiment we used float16 stable-diffusion-2-1-base[6] Rombach et al. (2022) and 50 DDRM steps, using the same hyperparameters. Due to the loss in PSNR due to using the VAE decoder, we focus Latent-PSC on high perceptual generation. We restore the images using the same model with the posterior sampler from Nested Diffusion Elata et al. (2024a), which applies 50 DDRM steps, each composed of 5 second order unconditioned diffusion sampling steps. We find this sampler to work better than $\Pi$GDM for this specific setup.

We used publicly available third party software for JPEG Wallace (1991), JPEG2000 Skodras et al. (2001), and BPG Bellard (2018). For HiFiC Mentzer et al. (2020), we trained our own model using the pytorch implementation publicly available on github[7]. We trained the models using the default parameters for each rate, and pruned networks that failed to converge to the desired rate. To provide results for ELIC He et al. (2022) and IPIC Xu et al. (2024) we used the official implementation for IPIC[8]. We used the Neural CompressionMuckley et al. (2021)[9] for MS-ILLM Muckley et al. (2023) results, and the unofficial pytorch implementation[10] of PerCo Careil et al. (2023).

FID Heusel et al. (2017) and KID Bińkowski et al. (2018) is measured using Pytorch Fidelity [11]. For FID on the $256 \times 256$ ImageNet images we used 50 random crops of size $128 \times 128$ inspired by Mentzer et al. (2020). For KID on experiments on the CLIC Toderici et al. (2020) and DIV2K Agustsson & Timofte (2017)

---

[5]https://github.com/bamler-lab/constriction

[6]https://huggingface.co/stabilityai/stable-diffusion-2-base

[7]https://github.com/Justin-Tan/high-fidelity-generative-compression

[8]https://github.com/tongdaxu/Idempotence-and-Perceptual-Image-Compression

[9]https://github.com/facebookresearch/NeuralCompression/tree/main/projects/illm

[10]https://github.com/Nikolai10/PerCo

[11]https://github.com/toshas/torch-fidelity

validation datasets, we used $64 \times 64$ patches from the $512 \times 512$ images, as a sufficiently large reference for high-quality images (The datasets are quite small).

The top image in Fig. 1 is taken from the Kodak Franzen (1999) Dataset.

## A.1 PSC Pseudo-Code

```
from utilities import posterior_sampler, restoration_function, entropy_encode,     1
    entropy_decode
                                                                                    2
def SelectNewRows(H, y, r, shape, s=None):                                          3
    c, h, w = shape                                                                 4
    s = s if s is not None else (r * 4) // 3                                        5
    noise = torch.randn((s, c, h, w))                                               6
    samples = posterior_sampler(noise, H, y)                                        7
    samples = samples.reshape(s, -1)                                                8
    samples = samples - samples.mean(0, keepdim=True)                               9
    new_rows = torch.linalg.svd(samples, full_matrices=False)[-1][:r]              10
    return new_rows                                                                11
                                                                                   12
def PSC_compress(image, N, r)                                                      13
    c, h, w = image.shape                                                          14
    H = torch.zeros((0, c * h * w))          # Empty sensing matrix               15
    y = H @ image.reshape((-1, 1))           # Empty measurements                 16
    compressed_representation = y.clone()                                         17
                                                                                   18
    for n in range(N):                                                            19
        new_rows = SelectNewRows(H, y, r, (c, h, w))                              20
        H = torch.cat([H, new_rows])                                              21
        y = torch.cat([y, new_rows @ image.reshape((-1, 1)])                      22
                                                                                   23
        compressed_representation = y.to(torch.float8_e4m3fn) # Quantize          24
        y = compressed_representation.to(torch.float32)                           25
                                                                                   26
    return entropy_encode(compressed_representation)                              27
                                                                                   28
def PSC_decompress(compressed_representation, N, r)                               29
    compressed_representation = entropy_decode(compressed_representation)         30
    c, h, w = image.shape                                                         31
    H = torch.zeros((0, c * h * w))          # Empty sensing matrix               32
    y = H @ image.reshape((-1, 1))           # Empty measurements                 33
                                                                                   34
    for n in range(N):                                                            35
        new_rows = SelectNewRows(H, y, r, (c, h, w))                              36
        H = torch.cat([H, new_rows])                                              37
        y = compressed_representation[:(n*r + r)].to(torch.float32)               38
                                                                                   39
    return restoration_function(H, y)                                             40
```

Our complete code is available at `https://github.com/noamelata/PSC`.

Latent-PSC follows the diagram in Fig. 7, encoding the input image with the VAE encoder before compression and decoding the decompressed output with the VAE decoder. Also, as the diffusion model used is text-conditioned, the text prompt is given to the diffusion model inside `posterior_sampler` and `restoration_function`. The text is also begin compressed and appended to the compressed representation.

## A.2 Computational Requirements

The computational requirements for PSC are listed in Tab. 1, with rate and time for both compression and decompression. Because PSC is a progressive compression algorithm, runtime increases with algorithm iterations, which correlate linearly with the compression bit-rate. The memory requirements are 12461MB for PSC 256 and 5719MB for Latent-PSC 512. These evaluation have been conducted on a single Nvidia A100 GPU.

Table 1: Computational requirements for PSC - Perception 256 (top) and Latent-PSC 512 (bottom).

| Iterations | 16 | 32 | 64 | 128 | |
|---|---|---|---|---|---|
| Rate (BPP) | 0.0097 | 0.0193 | 0.0387 | 0.0773 | |
| Compression time (min.) | 3:25 | 6:55 | 14:59 | 37:57 | |
| Decompression time (min.) | 3:49 | 7:20 | 15:31 | 38:21 | |

| Iterations | 16 | 32 | 64 | 128 | 256 |
|---|---|---|---|---|---|
| Rate (BPP) | 0.0065 | 0.0114 | 0.0211 | 0.0405 | 0.0795 |
| Compression time (min.) | 2:54 | 5:38 | 11:07 | 22:13 | 45:36 |
| Decompression time (min.) | 3:26 | 6:13 | 11:48 | 23:05 | 46:41 |

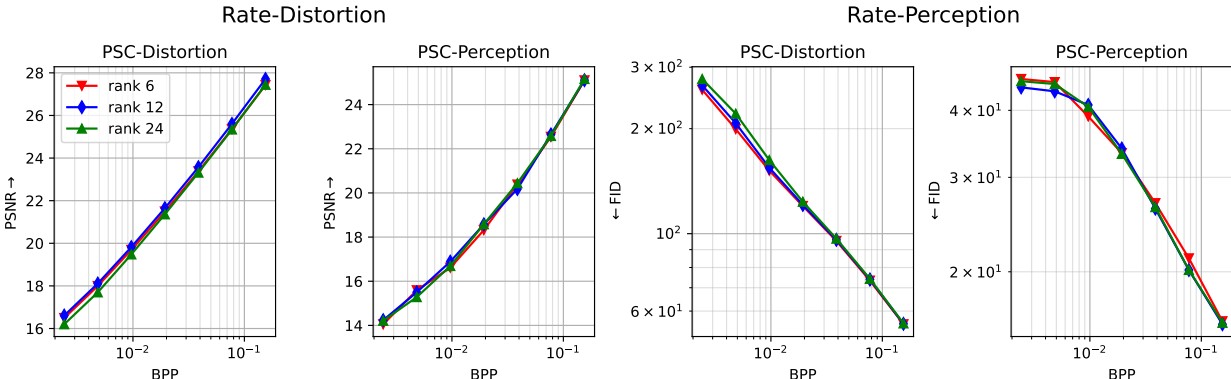

Figure 8: **Rate-Distortion (top) and Rate-Perception (bottom) curves for ImageNet256 compression, using PSC-Distortion (left) and PSC-Perception (right).** Distortion is measured as average PSNR of images for the same desired rate or specified compression quality, while Perception (image quality) is measured by FID.

## B  Effect of measurement rank

We repeat the ImageNet experiment with different values of the hyperparameter $r$, which determines how adaptive our algorithm would be. We modify the number of samples generated at each iteration $s$ accordingly to account for the rank required by the empirical covariance matrix. Based on the original implementation of AdaSense Elata et al. (2024b), we expect performance to improve the lower the value of $r$ is. In the results, demonstrated in Figure 8, the variation of the rank seems to have only a marginal effect, even for low rates. We conclude that PSC is not sensitive to this parameter, and $r$ can be tuned according to the system's hardware (namely, maximum available batch size).

## C  Additional Latent-PSC ablations

Figure 9 shows an additional comparison of Latent-PSC to PerCo Careil et al. (2023) and MS-ILLM Muckley et al. (2023), similar to Fig. 11. Figure 10 demonstrates the progressive nature of PSC – as more rows of $H$ are accumulated, the posterior distribution converges with the input. This comes at the cost of a larger compressed representation and higher bit-rates.

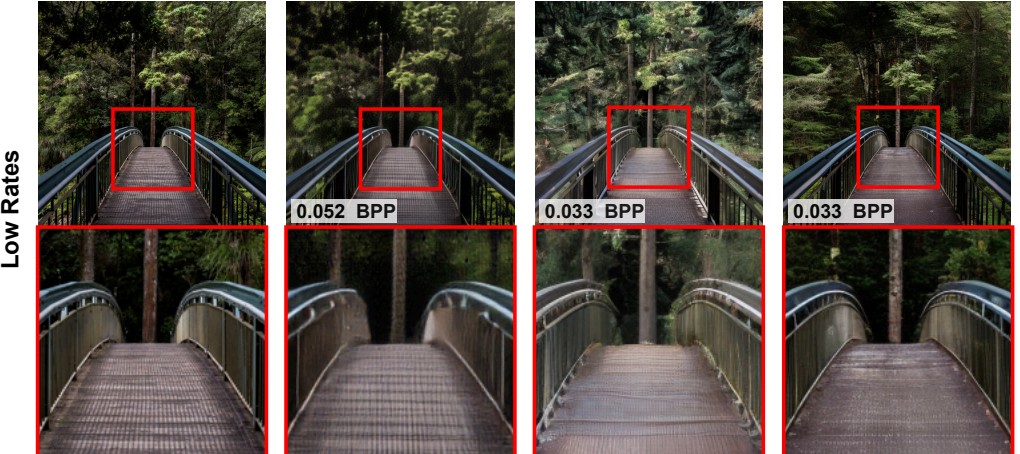

Figure 9: **Additional comparison of Latent-PSC to leading methods.** Zoom-in view is shown below each image

To quantify the effects of different rates and use of textual prompts, we evaluate Latent-PSC on $512 \times 512$ images from the MSCOCO Lin et al. (2014) dataset, which includes textual descriptions for each image. We compress the textual description assuming 6 bits per character, with no entropy encoding. Figure 11 shows decompressed samples using Latent-PSC with different rates, demonstrating good semantic similarity to the originals and high perceptual quality.

### C.1 Effect of Caption on Latent-PSC

Figure 12 illustrates the impact of using a captioning model to obtain the textual representation. In this experiment, the captions generated by BLIP Li et al. (2022) achieved comparable or superior results to human annotated description from the dataset. However, omitting the prompt causes some degradation of quality.

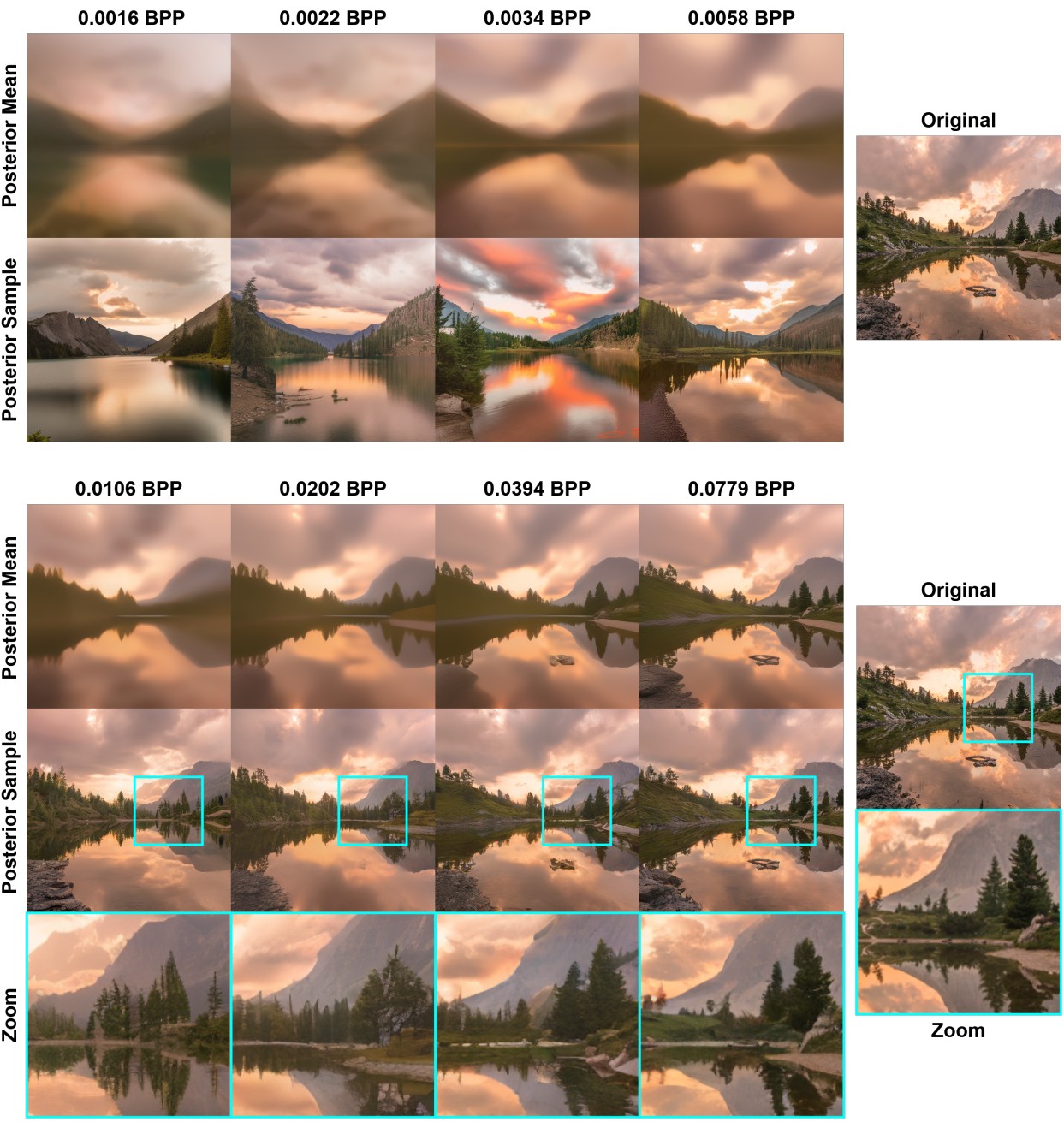

Figure 10: **An example of the progression of PSC with growing bit-rates from left to right (both rows).** The posterior mean (top of each row) becomes sharper as more information about the original image is accumulated. The posterior samples also converge to the original while always being of high photorealism. A zoom-in view is shown below the bottom row of images to highlight fine differences

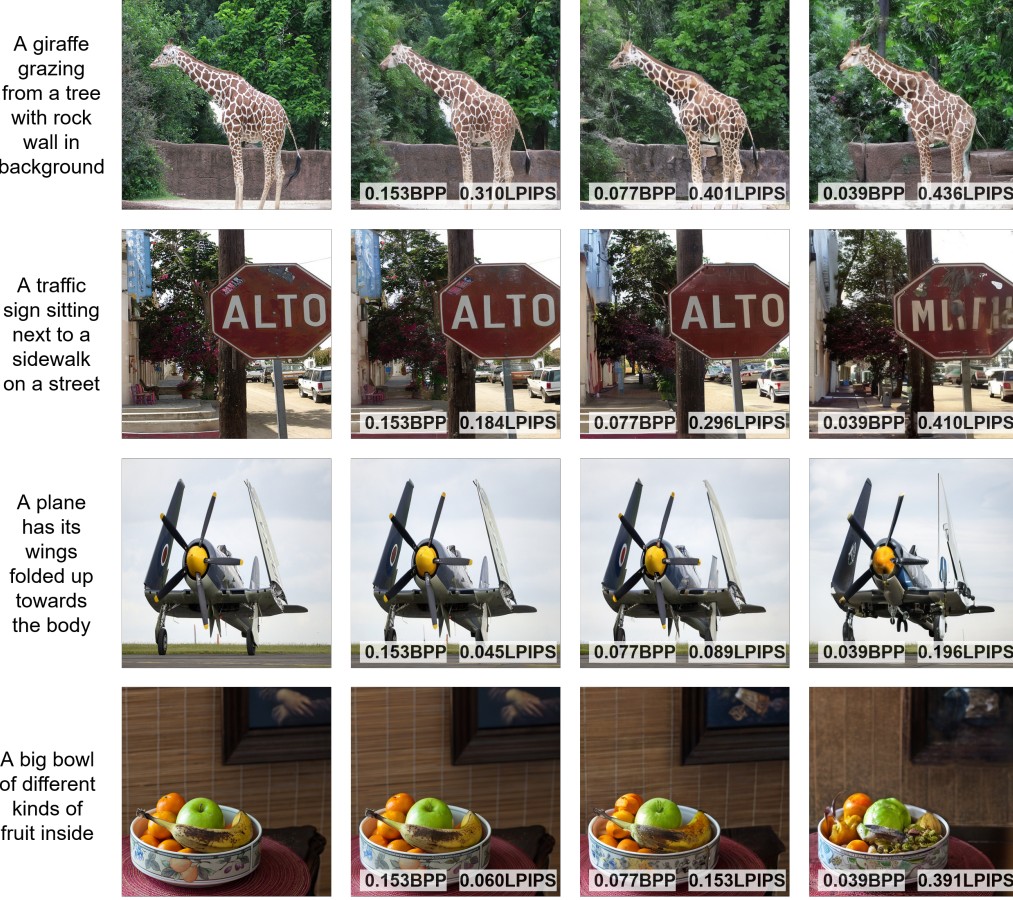

Figure 11: **Qualitative examples of Latent-PSC with Stable Diffusion.** For each image and corresponding text, several results for different bit-rates are shown. BPP and LPIPS are reported.

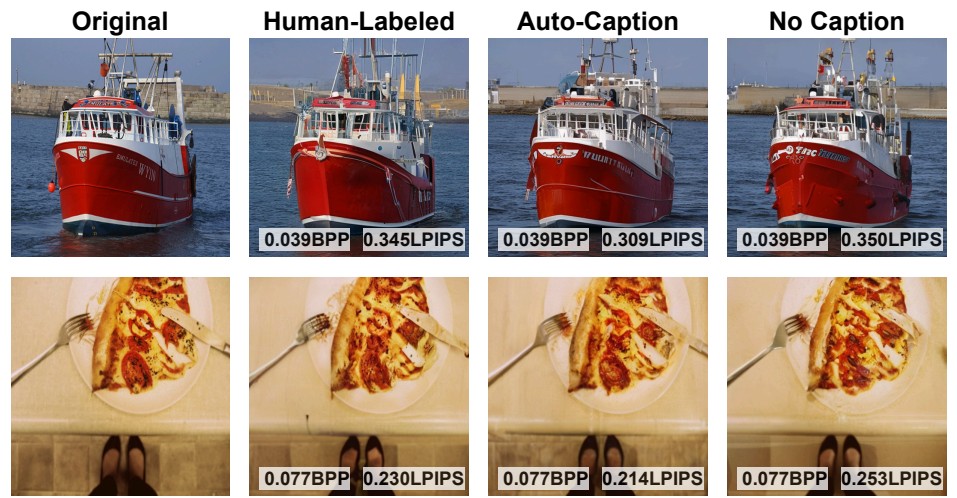

Figure 12: **Qualitative examples of Latent-PSC with various prompt configurations.** For each image we compare compression results with human annotated textual description, auto-captioning using a model, and using no caption.