# OpenReview forum: "PSC: Posterior Sampling-Based Compression"
_TMLR — Accepted by TMLR_

### Review · Reviewer_kacb · 2025-07-02

**Summary Of Contributions:**

The authors propose an algorithm that leverages diffusion as a prior to sample the posterior of a compressed image, which they coin Posterior Sampling–based Compression (PSC). Their work is inspired by AdaSense (an adaptive compressed sensing algorithm), but focuses on image-specific compression transformations instead of linear transformations. They introduce two main algorithms— the PSC encoder (Algorithm 1) and the PSC decoder (Algorithm 2) both of which leverage the posterior distribution. The encoder progressively expands the sensing matrix to encode compressed measurements y, and the decoder then reconstructs the true signal x from those measurements. To guarantee deterministic outputs when sampling from the posterior in both algorithms, a fixed random seed must be used. The authors demonstrate their method using both pixel-diffusion priors and latent-diffusion techniques with established posterior-sampling algorithms. Across a wide range of bits-per-pixel (BPP) settings, PSC outperforms strong contemporary baselines in both latent and pixel diffusion scenarios. The main contribhution of this work is strong experimental results.

**Audience:**

Yes

**Broader Impact Concerns:**

Not Available

**Claims And Evidence:**

Yes

**Requested Changes:**

1. In the Implementation section, add a link to the publicly available code repository and list the necessary computational requirements.
2. Replace the “advantages of using the proposed method” heading with “novelty of the work.”
3. Revise or remove Figure 2—Algorithm 1 is sufficient. If you keep it, add a sentence in the figure caption or main text explaining how Algorithm 1 differs from AdaSense (e.g. by introducing an additional sensing matrix H in the first few lines for conditional sensing).

**Strengths And Weaknesses:**

Strengths:

1. Strong empirical results: The proposed algorithm yields higher-fidelity reconstructions with fewer bits per pixel for pixel diffusion priors, outperforming strong contemporary baselines.
2. Simple, elegant implementation: The algorithm is easy to implement given adequate computational resources, and the authors provide in-depth implementation details in Section 3.1.

Weaknesses:

1. Aspects of the writing:
   a. In “To summarize, PSC offers the following advantages,” the last two bullet points merely reflect the use of a diffusion model as a prior, rather than novel contributions. Generative models as priors for inverse problems predate this work.
   b. In Figure 2, “H” is not labeled, and it is difficult to understand the added benefit of this diagram since Algorithm 1 already explains the concept clearly.
   c. The computational requirements should be specified in the Experimental Details section.
   d. In Figure 5 (Rate–Perception), it is hard to distinguish the exact performance difference between MS-ILLM and the proposed method.
2. Latent PSC evaluation: In the Latent PSC section, there appears to be no clear benefit of the proposed method over Perco, but is interesting that proposed alogrithm works for both pixel and latent diffusion models.
3. Code availability: The manuscript does not mention plans to publicly release the code or include it in the supplementary materials.

---

> ### Author Response · Authors · 2025-08-06
>
> Thank you for your comprehensive review and constructive feedback. We address each concern below:
>
> **Writing:**
>
>    **a) Bullet Points** We acknowledge your point that the last two bullet points reflect established concepts rather than novel contributions. These points were intended to highlight the specific advantages of our approach within the established category of diffusion model-based priors for inverse problems.
>
>    **b) Figure 2.** Please see below.
>
>    **c) Computational Requirements** We will add detailed computational requirements to the Experimental Details section, including specific hardware specifications, memory usage, and runtime comparisons with baseline methods.
>
>    **d)** We will improve Figure 5 by using different markers/colors to better distinguish the performance differences between MS-ILLM and our proposed method.
>
>
> **Latent Evaluation:** You correctly note that our method shows comparable rather than superior performance to Perco in the latent space evaluation. We view this comparability as a significant achievement given the fundamental differences in our approaches: our method operates zero-shot without requiring fine-tuning, while Perco benefits from extensive optimization specifically designed for neural compression. The fact that our general-purpose approach achieves comparable quality demonstrates its robustness and potential for broader applicability. We will clarify this distinction more explicitly in the revised manuscript.
>
> **Code Availability:** We commit to publicly releasing our code upon publication. We will include the implementation in the supplementary materials of the revision.
>
> **Replacing Advantages with Novelty:**  The bullet points in question were meant to summarize the benefits a practitioner would gain by adopting our method, including the zero-shot capability that diffusion-based priors enable. We would propose that clearly articulating these advantages serves an important purpose for readers who may understand our technical contributions but need guidance on when and why to adopt our method over existing approaches. However, if the reviewer maintains that focusing on novelty would better serve the manuscript, we are open to revising the section accordingly.
>
> **Figure 2:** The figure represents the AdaSense approach as applied for compression, and the only minor difference is the application of quantization. We had hoped this would help clarify the approach, but we agree that both are not necessary. As such, we will remove Figure 2 unless other reviewers express concerns about this change.

---

### Review · Reviewer_rszy · 2025-07-28

**Summary Of Contributions:**

Summary Of Contributions
The paper introduces Posterior Sampling-based Compression (PSC), a zero-shot image compression framework that utilizes a pre-trained diffusion model as its sole neural network component, eliminating the need for task-specific training. The method is inspired by transform coding but uniquely constructs an image-specific transform matrix progressively. The core innovation is a synchronized process where the decoder can independently reconstruct the exact same adaptive transform as the encoder by leveraging the transmitted quantized measurements and a shared random seed.

**Audience:**

Yes

**Claims And Evidence:**

Yes

**Requested Changes:**

1The paper must include a clear, quantitative analysis of the computational complexity in the main text. This should include metrics like encoding/decoding time on specific hardware and the number of function evaluations (NFEs) required at different rates, directly comparing these figures with the baseline methods.
2The authors should revise the methodology section to be more self-contained. This involves providing a higher-level intuition behind the AdaSense algorithm and explicitly explaining why minimizing the posterior covariance (i.e., uncertainty) is an effective proxy for optimizing rate-distortion performance in this context.
3The experimental section should be expanded to include more comparisons with other relevant and state-of-the-art methods.

**Strengths And Weaknesses:**

Strengths:
1The method's primary strength is its zero-shot nature, leveraging publicly available, pre-trained diffusion models without any additional training for the compression task.
2PSC provides precise, instance-level control over the compression rate and allows the user to decode for either low distortion or high perceptual quality from a single compressed representation.
3Despite its training-free approach, PSC achieves rate-distortion and rate-perception performance comparable to established training-based methods like HiFiC, particularly in low-bitrate scenarios.
Weaknesses:
1The most significant limitation is the substantial computational overhead. The method requires an order of 10,000 NFEs for both compression and decompression of a single image, making it impractical for many real-world applications. This cost was not adequately quantified or discussed in the main paper.
2The approach is a clever integration of existing technologies—namely diffusion-based posterior samplers (e.g., DDRM) and the AdaSense algorithm—which led some reviewers to question its fundamental novelty.
3The paper's reliance on the relatively obscure AdaSense algorithm made the methodology difficult to follow for readers unfamiliar with it.
4The evaluation was missing direct comparisons to some relevant state-of-the-art flexible compression methods. Furthermore, the lack of analysis on encoding/decoding time was a major omission.

---

### Review · Reviewer_n8gb · 2025-07-30

**Summary Of Contributions:**

This paper investigates the use of pretrained diffusion models for the purpose of image compression. The work builds on the recent AdaSense method, which provides a paradigm for using zero-shot posterior diffusion sampling for linear inverse problems with pretrained unconditional diffusion models. This method iteratively builds encoding coefficients $y$ and linear encoding transform $H$ for an observed image $x$ by drawing several samples from $p(x | y)$ under the assumption that $y = Hx$, taking the PCA of these samples, and adding the top PCA coefficients to $y$ and the corresponding PCA vectors to $H$. However, AdaSense cannot be used directly for compression because it requires the matrix $H$ to decode $y$, and storing $H$ requires a large amount of memory. This work provides a method for reconstructing $x$ using only the compressed measurements $y$, thereby enabling image compression. This is done by fixing the random seed of the diffusion model during both encoding and decoding and re-calculating the $H$ matrix iteratively during the decoding using parallel posterior diffusion samples with the same seed. Once $H$ is obtained from $y$, the user is able to select diffusion zero-shot posterior sampling algorithms to get the final reconstruction. Experiments apply the proposed method to reconstruct ImageNet 256x256 samples using an unconditional pixel space diffusion and reconstruction using a latent space text-conditioned diffusion. The ImageNet experiments present two versions that use different zero-shot posterior sampling: one with low distortion at low BPP compared to related methods, and one with high perceptual quality at low BPP compared to related methods. The latent diffusion experiments cannot provide low distortion due to VAE reconstruction, but can demonstrate high perceptual quality at low BPP compared to related methods.

**Audience:**

Yes

**Broader Impact Concerns:**

Broader impacts are not discussed, but that does not impact my evaluation of the paper.

**Claims And Evidence:**

Yes

**Requested Changes:**

My main requested change is that a thorough report on NFE and runtime (and perhaps other computational requirements like GPU memory use) of the method is added. This provides important context and also provides a baseline for future work on diffusion-based compression to compare against.

I have a few questions that would clarify the work.

* Can you elaborate on how $y$ and $H$ are selected at the very beginning of the compression stage when they are designated as empty? Does this mean that unconditional samples are drawn from the diffusion model $p(x)$ and used to provide initial PCA directions? It seems like this very first step does not use any information from the input image, is that right? Can this step be "skipped" by just memorizing $H$ from the PCA of unconditional samples?
* Can you elaborate about the practical use of encoding with high perceptual quality? Are there situations where this is important?

**Strengths And Weaknesses:**

*Strengths*
* The proposed encoding method provides a very straightforward and elegant way to extend AdaSense to an image compression method by seeding posterior sampling in the same way both during encoding and decoding, which allows image reconstruction from only the measurement code (at the expense of re-performing computationally intensive diffusion during decoding).
* The overall goal of the paper is interesting and novel. As the authors note, image compression has been discussed even in very early work on diffusion modeling, but this method (to my knowledge) provides the most concrete and convincing practical demonstrating of diffusion-based compression.
* The experiments show impressive results in terms of low-BPP reconstruction using both distortion (PSNR) metrics based on the DDRM algorithm and perceptual quality based on the $\Pi$GDM algorithm.

*Weaknesses*
* The primary weakness of the proposed algorithm is the high computational cost of the algorithm. Thousands of NFEs with large diffusion models are needed to encode and reconstruct a single image. While the aim of this work seems to be more of a proof of concept than a practical tool, it is still a major limitation. The cost is only discussed briefly in the paper and a thorough breakdown of NFEs and runtimes for different situations would provide important context for the reader.
* The method is limited in its reconstruction abilities when the diffusion model is a latent space model. It is well-known that even the most advanced autoencoders used in latent diffusion introduce significant reconstruction error for small details and high frequency patterns. Since autoencoder reconstruction quality provides the upper bound for reconstruction from the proposed method when the diffusion model is latent, the proposed method is not suitable for low-distortion encoding when performed in the latent space. Given that low-distortion encoding is probably the most important aspect of compression, the fact the most recent diffusion models are in the latent space, and the extreme cost of pixel space diffusion for high-resolution images, there are a lot of practical obstacles for this method to achieve low-distortion encoding with off-the-shelf diffusion models.

---

> ### Author Response · Authors · 2025-08-06
>
> Thank you for your comprehensive review and constructive feedback. We address each concern below:
>
> **Computational Cost:** We acknowledge that computational cost represents the primary limitation of our proposed method. As the reviewer correctly notes, our work serves as an initial proof of concept demonstrating the feasibility of our method, with the expectation that future research will develop more computationally efficient approaches. We will include a breakdown of the computational requirements across different bit rates in the revised version.
>
> **Latent Space Compression:** We agree with the reviewer's assessment that latent space diffusion imposes fundamental limitations on low-distortion encoding due to the reconstruction errors inherent in the decoder. This limitation is indeed shared with other recent latent space compression methods such as PerCo. We believe this limitation can be partially alleviated by replacing the existing VAE decoder trained using GAN-based losses for perceptual reconstruction with one trained for MMSE reconstruction. This option may be an interesting avenue for future work.
>
> **Initial Selection of $\boldsymbol{H}$:** The reviewer correctly identifies that the initial step can be optimized by precomputing $\boldsymbol{H}$ based on the prior distribution $p(\mathbf{x})$ rather than generating samples with the model. We omit this optimization for two key reasons: First, implementing a special case for the initial iteration would introduce additional algorithmic complexity. Given that our work serves as an initial exploration rather than an optimization-focused implementation, we prioritized algorithmic clarity and simplicity. Second, precomputing $\boldsymbol{H}$ becomes significantly more complex in text-conditional settings (as employed in our latent space experiments), and we sought to maintain consistent methodology across both experimental configurations. We will discuss this optimization and its potential implementation in the revised manuscript.
>
> **High Perceptual Quality:** High perceptual quality compression offers significant advantages in applications where semantic fidelity and visual appeal take precedence over pixel-level accuracy. High perceptual quality reconstruction preserves semantic content and overall image aesthetics while producing results that appear uncompressed to human observers. In contrast, low-distortion approaches in low bit-rate scenarios often yield heavily blurred reconstructions that, despite favorable PSNR scores, may not be perceived as valid or useful images by end users. This distinction becomes particularly important as compression ratios increase and the trade-off between perceptual quality and distortion metrics becomes more pronounced.

---

### Review · Reviewer_RYVp · 2025-08-22

**Summary Of Contributions:**

This paper proposes an image compression method which utilizes a diffusion model to generate the linear encoding matrix. During encoding, the diffusion model is used to build this matrix by focusing on the principal components, thereby achieving a compressed representation through component weights, which can then be quantized and entropy encoded. During decompression, the compressed representation along with the initial diffusion seed can be used to reconstruct this matrix and subsequently a form of the original image. Additionally, the diffusion networks are zero-shot, establishing a training-free approach, and result in a more generalizable approach that prior works. The authors demonstrate strong results on ImageNet samples compared to baselines, and on both CLIC and DIV2K for their latent diffusion-based version.

**Audience:**

Yes

**Broader Impact Concerns:**

No concern.

**Claims And Evidence:**

Yes

**Requested Changes:**

Changes:

- Could the authors add a discussion on how the diffusion model is incorporated into the posterior sampling loop? In the pseudocode, what does `posterior_sampler ( noise , H , y)` look like? Most networks operate on only the noise and label (integer class or text), which is different from the compressed vector `y`.

- Sampling configuration and sensitivity to sampling hyperparameters is also necessary to understand the evaluations and for reproducibility. As it stands, this part of the method is described as a black-box. Additionally, are the results sensitive to the ODE solver, sampling timesteps, network / network size, etc.

- A clearer discussion on how the encoder and decoder networks are linked would also be helpful. Can these be different networks? If so, then that adds another level of generality, whereas if not, then the network description would also be needed for reconstruction.

- Are there any limitations with image resolution? Most diffusion models have trouble generating at resolutions OOD from the training cases.

Questions:

- For the latent diffusion version, is the text description included in the BPC calculation, since this information must also be transmitted? And is the latent diffusion version sensitive to the quality and length of the text description?

- Could the authors provide some motivation as to why this problem is interesting from a practical perspective? Despite the efficiency gains in diffusion models, the computational cost is likely not worth the small quality increase when compared with BPG and JPEG2000.

- Does the compressed version `y` contain semantic information similar to a CLIP embedding, independent of the compression process? In such a case, a dedicated diffusion model could be trained to decode `y` directly, without the expectation of strong perceptual similarity but instead focusing on semantic similarity.

**Strengths And Weaknesses:**

Strengths:
- The paper is clear and well written, using a reasonable approach to including diffusion models in compression and decompression.
- Multiple datasets, resolutions, and baselines are considered, demonstrating better or comparable results.
- The inclusion of a latent diffusion-based version utilizing text-to-image models is interesting, where the image description acts as part of the compressed representation.

Weaknesses:
- The proposed approach is not well motivated, especially considering the high computational cost involved in the compression and decompression process.
- The compression approach appears to be largely semantic and layout in nature, resulting in specific loss of details as shown in the qualitative examples.
- While the compression and decompression outer-loops are well described, the incorporation with the diffusion model is not.

---

> ### Author Response · Authors · 2025-08-28
>
> Thank you for your comprehensive review and constructive feedback. We address each concern below:
>
> **Motivation and Practical Significance** Our work introduces a novel semantic compression paradigm that exploits learned representations from diffusion models. While computational efficiency remains a current limitation, this proof-of-concept establishes new foundations. The zero-shot nature requires no task-specific training while demonstrating flexibility across datasets, rate, and desired perceptual quality. As diffusion models and posterior samplers become more efficient, the computational costs limiting practical deployment are expected to diminish significantly. We position this as foundational research that opens new directions rather than an immediately deployable solution.
>
> **Semantic Compression** The semantic nature of our approach is one of our method's strengths, as it focuses the compression on what the models have learned to be important for photorealistic images. At the same time, this means that we inherit the limitations of whichever diffusion model we utilize, resulting in specific loss of details in cases where the diffusion model does not perform optimally.
>
> **Description of the Diffusion Sampling** PSC can make use of any zero-shot posterior sampling algorithm with the same diffusion model. Specifically, we have used DDRM, $\Pi$GDM and the posterior sampler from Nested Diffusion to produce `posterior_sampler(noise , H , y)`. Each of these methods condition the reverse diffusion process on compressed measurements $\mathbf{y}$ through the linear operator $\boldsymbol{H}$. We will provide algorithmic descriptions in the revised manuscript.
>
> **Hyperparameters** Our experimental setup inherits most hyperparameters from the underlying diffusion models and posterior samplers. PSC introduces minimal additional parameters, such as the number of measurements added at each iteration, but we find that many of our hyperparameter choices work equally well.
>
> **Encoder and Decoder** The encoder and decoder networks must be identical to ensure deterministic reconstruction. This requires that both the encoder and decoder coordinate the network (equivalent to the coordination of the compression algorithm), as well as a random seed. Interestingly, the rate at which the image has been compressed can be determined at runtime on the decoder side without coordination.
>
> **Resolutions** Our approach inherits resolution limitations from the underlying diffusion models, as is the case in our experiments. Nevertheless, as a zero-shot method, PSC readily adapts to any diffusion architecture (including resolution-agnostic models) and is therefore not limited in resolution.
>
> **Textual Description** As the reviewer suggests, the textual description is included in our bits-per-pixel calculations. We have included a small ablation in appendix D on the effect of removing the textual description or making use of generated captions which are less accurate, and we find this only has a minor effect on the output. The relative size of the textual description is nearly negligible compared to the other information being transmitted, suggesting that it is worth including even for a small gain in performance.
>
> **Decompressing $\mathbf{y}$** The author raises a great point, a model can be trained directly to decompress $\mathbf{y}$, focusing either on image fidelity or on perceptual quality. In this work we have opted to focus wholly on a zero-shot implementation, which requires no additional training besides that of the original diffusion model.

---

### Decision · Action_Editor_Cu3k · 2025-09-05

**Recommendation:** Accept as is

**Audience:**

Yes

**Audience Explanation:**

All reviewers were positive about the paper. They acknowledged the methodological contribution as noteworthy, with performance metrics demonstrating competitive outcomes relative to existing techniques in the field. While the theoretical framework presents clear mathematical foundations that facilitate implementation for those with sufficient infrastructure, questions remain regarding the generalizability of experimental protocols and the granularity of technical specifications necessary for independent validation. The work's potential impact on the research community appears substantial, though practical deployment considerations and the completeness of methodological documentation warrant further attention to maximize its utility for practitioners seeking to build upon these findings. That being said, the paper can be accepted as is.

**Claims And Evidence:**

Yes

**Claims Explanation:**

The paper presents a compression framework that offers dynamic control during encoding, allowing users to adjust quality-size tradeoffs by iteratively adding measurements until target criteria are met. The system supports flexible decoding strategies where the same compressed representation can be reconstructed at different quality levels based on the chosen restoration method. The approach operates without training on specific datasets, leveraging diffusion model capabilities to handle diverse data modalities beyond images. Additionally, the architecture is designed to automatically incorporate improvements in generative modeling techniques without requiring system updates or retraining. All claims are supported by quantitative and qualitative results.